# Safe Return to Work for Domestic Workers in the Time of COVID-19

**Rupkatha Bardhan** * **, Traci Byrd and Julie Boyd**

Department of Occupational Safety and Health, Murray State University, Murray, KY 42071, USA;
tbyrd@murraystate.edu (T.B.); jboyd16@murraystate.edu (J.B.)
* Correspondence: rbardhan@murraystate.edu; Tel.: +1-270-809-3583; Fax: +1-270-809-3630

**Abstract:** Domestic workers including housecleaners, nannies, and caregivers are facing a challenging time in the era of COVID-19 exposure. Many domestic workers have lost their jobs worldwide. As businesses and organizations have started to reopen in full capacity, domestic workers are unsure of their future and whether they will be rehired by their employers. They have less protections from labor laws unlike other occupations and usually their employers/agencies do not provide training on safe practices for working in a home setting. There are gaps in understanding safety and health issues associated with precarious work for domestic workers. This review article has searched the literature on safe strategies for domestic workers to eliminate exposure and provides helpful suggestions for domestic workers to safely return to work. Employers or house owners can have a proper reopening plan when considering hiring or rehiring domestic workers. Domestics working in a home environment should use best practices to protect themselves and others from infectious diseases. Having open communication between employers and their domestic workers can go a long way. Implementing and following an effective working plan for both employers and their domestic workers will provide a path towards minimization of hazard and control of infectious diseases like COVID-19.

**Keywords:** domestic workers; domestic work; COVID-19; challenges; reopening plan; recommendations

## 1. Introduction

Domestic workers including house cleaners, nannies, babysitters, and caregivers can work either full time or part time in a home setting. Their common household jobs are not limited to cleaning, cooking, washing, and ironing clothes, and can also include taking care of children and elderly or sick member(s) of a family, lawn care, home security, driving for the family, and pet care [1]. They are often self-employed or employed through agencies. Domestic workers face a range of workplace issues including hazards and stress [2]. Occupational stress in domestic workforce can be attributed to a number of chronic factors like hostile or hazardous environments, low wages, inadequate access to medical care, physical demands of the occupation, exposure to infectious diseases while caring for sick elderly people, and exposure to toxic chemicals [2]. Chemicals commonly used for household cleaning and lawn care purposes are known to pose health risks and may cause skin rashes, burns, cough, asthma, cancer, cardiovascular diseases, and other toxicities [3,4]. Moreover, working in a home setting is not exactly considered a workplace and so the tasks accomplished by domestic workers are not considered work related. All these factors could negatively affect the physical and mental health of the domestic workers and may also lead to decline in productivity [1].

The challenges that domestic workers face in their job has escalated with COVID-19 situations. The virus that causes COVID-19 is SARS-CoV-2, which can be transmitted from human-to-human interactions [5]. Millions of domestic workers worldwide have lost their jobs or been sent away from their jobs due to lockdown and social distancing during the pandemic [6]. While some domestic workers are laid off from job with salary payments

provided by their employers, others are not so lucky. Although the situations of domestic workers might change for the better as communities begin to reopen, some employers can be still skeptical in hiring domestic workers. This article discusses the best practices for domestic work that will help both employers and domestic workers to get back to normal work conditions and work safely without risking the spread of infectious diseases.

## 2. Materials and Methods

The literature search was conducted using information on "Domestic workers, Domestic Work, Precarious Work, COVID-19, Safety return to work, Nannies, Caregivers, Babysitters" at PubMed, Google Scholar, Science Direct, Scopus publications, the web, media, news articles, and magazines, as well as Occupational Safety and Health (OSHA), the Center for Disease Control and Prevention (CDC), the National Institute for Occupational Safety and Health (NIOSH), and the United States Environmental Protection Agency (USEPA) websites. The literature search was conducted between June of 2020 to September of 2021. Very few peer-reviewed articles were identified related to the information on domestic workers' safe return to work after pandemic and, therefore, electronic articles, governmental websites, and news articles were also considered for this review. Information was gathered from these sites and the review was written on problems associated with domestic work encountered before and during COVID-19 and safe strategies for domestics to return to work.

## 3. Results and Discussion

The findings from the literature showed that the challenges domestic workers face in their work escalated with COVID-19. Domestic workers come from low economic and vulnerable backgrounds [7]. Their vulnerability in pre-pandemic situations was further escalated by the loss of jobs and uncertainty of rehire during the pandemic, causing mental stress and suffering [6,7]. The literature search was conducted to find out information on safe job practices for domestic workers to return to work. The information includes effective use of respirators and other personal protective equipment and safe guidance from governmental and other organizations.

### 3.1. Challenges Faced by Domestic Workers

According to Economic Policy Institute, around 2.2 million people work as domestic workers in the U.S. Most of them are women [8]. During the spread of SARS-CoV-2, many employers have been skeptical in hiring and keeping their domestic workers as they may fear domestic workers can spread the infectious virus by working in different homes. A survey conducted by the National Domestic Worker Alliance Executive Group reported that about 70% domestic workers (who were surveyed) have lost their jobs and wages [8]. Many employers did not pay wages to their domestic workers during the lockdown [9]. The situation might get worse as businesses and communities start to reopen, when domestic workers have very less chance of getting rehired due to the concern of spreading COVID-19 [8].

Domestic workers may come from a low-income background and are considered to be a vulnerable population with lower education levels and poor awareness of safety and health hazards associated with their work [10]. They do not receive proper training to deal with exposures as other workers get from their employers in a proper workplace setting. There are gaps in understanding health and safety risks and contributing risk factors associated with domestic work while working in a home-based environment [11]. Understanding those underlying risk factors associated with domestic work will help remediate the problem and provide ways to recommend 'best practices' for domestic workers.

Domestic workers are a predominantly vulnerable group with poor legal and social protections [12]. The Fair Labor Standards Act (FLSA) in the U.S was enacted in 1938 to provide minimum wage and overtime protections for workers, but excluded cooks, housekeepers, maids, and gardeners [13]. The FLSA law of 1938 established a minimum

wage of 25 cents per hour (which later increased to 30–40 cents per hour) for a standardized 44-h workweek. The federal minimum wage, which is contained in FLSA, for covered nonexempt employees is currently USD 7.25 per hour. Many states also have minimum wage laws. In cases where an employee is subject to both the state and federal minimum wage laws, the employee is entitled to the higher of the two minimum wages [13]. Congress expanded protection of domestic service workers in 1974. However, casual babysitters and domestic service workers employed to provide "companionship services" to elderly persons or persons with illnesses, injuries, or disabilities are not required to be paid the minimum wage or overtime pay [13]. Those who are self-employed domestic workers often work without the protection of formal employment contracts. In addition, domestic workers also face other unique challenges given their socioeconomic status, and in some cases, immigration status [8]. Women mostly work as domestic cleaners in the U.S and may be further disadvantaged if they are of Hispanic origin and undocumented [8]. Often, Hispanic workers are chosen to work in homes for low wages [7,8]. Economic and social deprivation together with uncertainties in employment conditions and workplace psychosocial stressors may adversely impact the mental health of women working as domestic cleaners [2,7].

When domestic workers are hired through agencies, they often bring their own cleaning supplies to clean homes. Most agencies are switching to more green and non-hazardous chemicals [14]; however, this may be not the typical case for all agencies. Some agencies might still prefer to use harsh chemicals as opposed to green chemicals, which may be costly or may not be as effective cleaners. However, self-employed domestic workers mostly use cleaning solvents provided by homeowners. They can be exposed to hazardous cleaning solvents provided by homeowners that they use to clean households [2]. These cleaning solvents can cause serious respiratory illness, skin damage, and even cancer after long-term use [2,14]. According to an article published by Peggie R. Smith in 2011, domestic workers experience higher rates of debilitating musculoskeletal disorders than any other occupational group in the United States including workers in coal mines and steel mills [15]. The high rate of musculoskeletal disorders reflects the type of work that these workers perform. Domestic workers are often scared to share their problems and concerns with their employer in fear of retaliation and loss of their job [2].

The Bureau of Labor Statistics (BLS) Jobs Report, released on 8 October 2021, shows the number of jobs added in September of 2021 in the U.S was below market expectations. This news release of the BLS presents statistics from two monthly surveys, and the household survey data showed that the unemployment rate fell by 0.4% to 4.8% in September 2021. The number of unemployed persons fell by 710,000 to 7.7 million. Unemployment rates for Black and Latino women decreased in September 2021. This was due to both an increase in employment and a decrease in labor force participation. The unemployment rates for Black and Latino adults continue to be higher compared to the rates for White and Asian adults in the U.S. People who have been unemployed long term (27 weeks or more) represented 34.5% of the total unemployed in September 2021 [16].

Each month, the U.S Bureau of Labor Statistics (BLS) releases an Economic Situation Summary with employment and other labor market data. However, domestic workers, along with other vulnerable workers, are often underrepresented in official data. There are currently estimated to be 2.2 million domestic workers throughout the United States, but this number does not encompass all who work on a cash basis [7]. The current status of those domestic workers returning to work is not documented in the United States Bureau of Labor Statistics (BLS). The BLS releases a quarterly report on the measurements compiled from the Quarterly Census Employment and Wages (QCEW), Business Employment Dynamics (BED), and Current Employment Statistics. Most of these reports exclude data from private homes, and agriculture jobs covered by unemployment. The sectors such as agricultural, private households, and positions held by self-employed workers are excluded from the data [16].

The National Domestic Workers Alliance (NDWA) conducted surveys and the report of the survey compares the indicators for domestic workers' joblessness, wages, housing security, and food security in the third quarter of 2021 (July, August, and September) versus the third quarter of 2020 (July, August, and September). The total number of fully completed surveys for September included 5616 respondents, for August there were 5840 completed surveys, and there were 4754 for July. While there has been some job recovery, domestic workers remain in a precarious economic situation. NDWA Labs' September Report shows that the percentage of jobless respondents in September was unchanged compared to August, and joblessness remains very high for Spanish-speaking domestic workers. In September 2021, 28% of domestic worker respondents were still out of work, much higher than the 9% who reported having no jobs before COVID-19 [17].

The National Domestic Workers Alliance (NDWA) report shows 28% of respondents were out of work on average in the third quarter (July, August, and September) of 2021, compared to 36% of respondents during the third quarter of 2020. In the third quarter of 2021, 47% of respondents faced housing insecurity, compared to 57% of respondents in the same quarter of 2020. The report includes that three out of four respondents faced some level of food insecurity in the third quarter of 2021. Out of the respondents, 77% of domestic workers experienced food insecurity in the third quarter of 2021 compared with 81% in the same quarter of 2020. An average of 86% of domestic worker respondents earned USD 15 or less per hour this quarter, a slight increase compared to 85% in the third quarter of 2020. Most domestic worker respondents, whether or not they had current work, looked for additional work in September [17].

### 3.2. Vulnerability of African Americans and Latino Women as Domestic Workers

Latino and African American workers are disproportionately represented in essential jobs that historically have poor work conditions [6]. The situation has got worse during a pandemic. The coronavirus pandemic is placing the nation's 2.2 million domestic workers—91.5% of whom are women—in a particularly precarious position [8]. There is a steep decline in domestic work leading to lack of jobs and loss of income for workers and their families. A lack of personal protective equipment for domestic workers adds to the health threat they face at work [18,19]. Domestic work is facing a long-term uncertainty. African American and Hispanic populations have a higher health risk to chronic and infectious diseases compared to the White population [20]. Domestic workers mostly do not have a health insurance coverage provided to them by their employers. The decline in employment for domestic workers represents a significant loss of income for these workers and their families [8].

While women of all races and ethnicities are overrepresented in the domestic employee workforce as shown in Figure 1, this overrepresentation is particularly pronounced for Hispanic and Black women [8]. The pie chart in Figure 1 is based on data from the Economic Policy Analysis of Current Populations survey and basic monthly microdata from the U.S Census Bureau [7]. House cleaners constitute the domestic worker occupation with the highest share of Hispanic workers (61.5%), while agency-based home care aides constitute the domestic worker occupation with the highest share of Black, non-Hispanic workers (30.3%). The statistics on pie Figure 1 show that mostly African American and Hispanic women in the U.S work in domestic sectors. Domestic workers are more likely than other workers to have been born outside the U.S (pie Figure 1) [7]. Immigrant women are often hired to do domestic work for low wages, and they face challenges to protest against their employers' wrong doings for fear of losing jobs [21,22]. Lack of training and lack of agency support might make it worse for domestics to work safely in an employer's home [15].

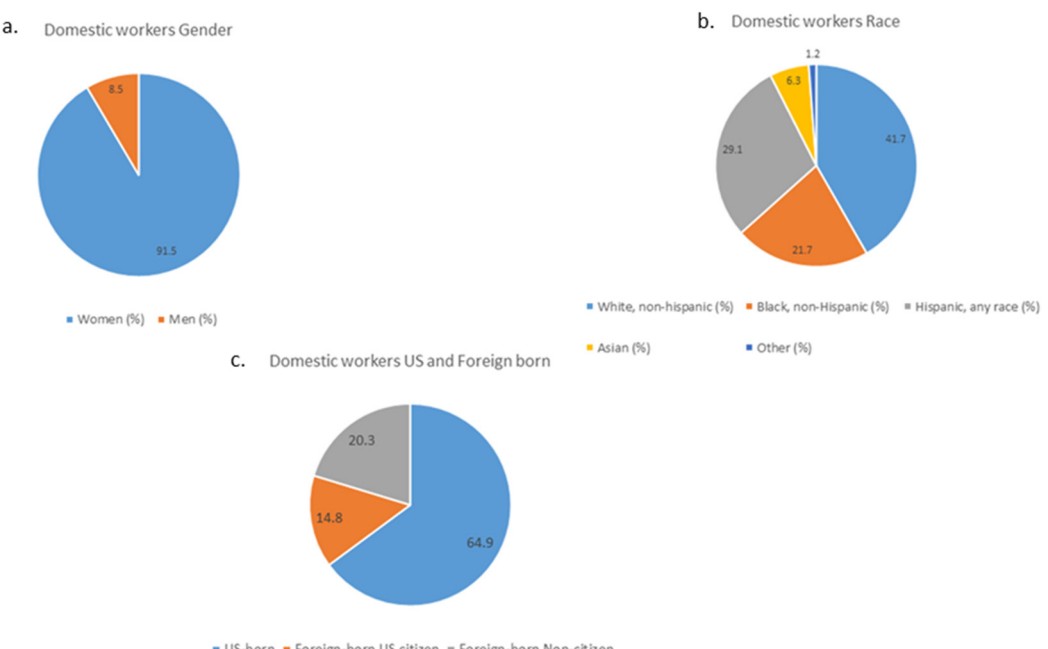

**Figure 1.** Pie charts showing domestic worker information from a population survey. (**a**) The chart represents shares of domestic workers in different occupations by gender in 2019. (**b**) The chart represents shares of domestic workers in different occupations by race/ethnicity in 2019. (**c**) The chart represents shares of domestic workers in different occupations by nativity in 2019.

There is no direct connection between sex and racial vulnerability and safe return to work. However, gender and racial vulnerability help to explain the challenges that the domestic workers face in their everyday work. These challenges further increased the vulnerability of the group during COVID-19 with loss of jobs, lack of information or misinformation and uncertainty of future [15,23]. This group of workers has less labor protection than usual workplace employees. Mostly, immigrant domestic workers struggle to support their families with poor income, insecure jobs, and absence of extended families' financial support [7]. Understanding the challenges that domestic workers face will help gather the support they need from their employers and the society to safely return to work.

Restrictions imposed in response to diseases transmissions (e.g., stay-at-home measures for COVID-19) may directly exacerbate the social support received by the foreign domestic workers and their ability to access health-related information. There are also concerns about their ability to appraise and evaluate information related to communicable diseases at a time when misinformation is being disseminated through social media. Language and cultural barriers are important issues that need to be addressed to ensure that foreign domestic workers can follow public health recommendations [24].

*3.3. Challenges Faced by Domestic Workers in Asian Countries*

Latino and African American workers are not alone in their plight. Nearly a decade after the adoption of the Domestic Workers Convention, domestic workers across the globe remain vulnerable due to the precarious nature of the demands of their jobs. In 2021, the International Labor Organization (ILO) reported that there are "at least 76.5 million domestic workers aged 15 and over, amounting to around one in 25 people employed worldwide" [6,25]. Below is a global snapshot of the countries making substantial employment contributions to the domestic worker population:

- Asian and Pacific Region = 38.3 million;
- China = 22 million;
- India = 4.8 million;
- Philippines = 2 million;
- Bangladesh = 1.5 million; and

- Indonesia = 1.2 million [25].

Domestic workers are considered "essential" as they allow people to work outside the home, thereby keeping global markets working [26]. ILO Director General Guy Ryder said "women comprise more than 80% of those working in the informal sector, which makes them more prone to exploitation and abuse" [27]. Worldwide, an extremely high percentage of domestic workers, both formal and informal (i.e., "performed outside of labour regulations and social protections" [26]), are excluded from national labor laws. This exclusion limits workers' social protections such as "working hour limitations and entitlement to weekly rest" [25]. Domestic workers earn some of the lowest wages in the world and work under some of the poorest conditions. These issues have worsened during the COVID-19 pandemic.

During the COVID-19 era, domestic workers have either continued working despite the pandemic or been relieved of their duties without pay. Those who have continued working have seen their workloads increase in extraordinary ways, including longer working hours and increased tasks (i.e., extra cooking, cleaning, etc.) due to families staying at home due to lockdown orders. However, they have not experienced an increase in pay and often are not allowed to take their regularly scheduled day off due to official stay at home recommendations [28]. As for those who once resided with their employer and have been relieved of their duties, they now find themselves both jobless and homeless, leaving them more vulnerable to physical and mental health issues. These issues can and often do lead to a "greater risk of falling into situations of trafficking or exploitation as they try to survive" [28].

Work-related transmission is considerable in early COVID-19 outbreaks, and the elevated risk of infection was not limited to health care workers. A study published in the PLOS ONE journal in May 2020 showed that among domestic workers in six Asian Countries, including Hong Kong, Japan, Singapore, Taiwan, Thailand, and Vietnam, possible work-related COVID-19 transmissions occurred [29].

The pandemic posed a serious threat to domestic migrant workers stranded in India due to the lockdown. Many these workers were left with no economic support, no food, and in many cases nowhere to live [30]. A study published in 2021 reported that migrant workers in Vietnam suffered from poor health and low occupational safety, fear of job loss and income cut, poor housing and living conditions, and limited access to public services. Most migrant workers in the study were female (65.2%), aged between 18 and 29 years old (66.8%), and had high school or higher education level qualifications. This study explored the impact of COVID-19 on migrant workers in Vietnam, using a cumulative risk assessment (CRA) framework, which comprises four domains (workplace, environment, individual, and community) [31].

Migrant workers have been one of the most vulnerable population groups during the COVID-19. On 30 January 2020, 7818 cases had been confirmed globally, and approximately 98.9% of the cases were in the Greater China Region, including Macao (Special Administrative Regions or SAR), Hong Kong (SAR), and Taiwan. A study investigated knowledge and awareness of COVID-19 among Indonesian migrant workers in Macao, Hong Kong, and Taiwan. One-third of the study participants reported receiving hoax, fake news, and incorrect information and obtained information from unverified sources. Participants with senior high school or higher education had a greater knowledge of COVID-19. The study recommended digital literacy content added to public health campaigns [32].

### 3.4. NIOSH Hierarchy of Controls for COVID-19

Ensuring the health and safety of workers are in place is essential, as we prepare for complete normalcy in businesses and services in coming months. Safety protections must align with the "hierarchy of controls" that favors more protective elimination, substitution, and engineering controls over less protective administrative controls and personal protective equipment [33]. To protect workers from SARS-CoV-2, administrative controls and personal protective equipment (PPE) will also be needed, as some people are still not

vaccinated and vaccines may not protect people from new variants of COVID-19 [34,35]. Personal protective equipment (PPE) is essential to protect workers from others who may be exposed. Controls should address three ways of virus transmission as illustrated by Figure 2. The Figure 2 chart is based on the concept of modes of transmission of SARS-CoV-2 from the Center for Disease Control and Prevention (CDC), World Health Organization (WHO), and Delikhoon et al. [36–38]. The three ways are as follows: (1) via large droplets expelled by infected persons who cough, sneeze, etc.; (2) via virus particles that can collect on surfaces that are then touched; and (3) via airborne aerosol micro droplets that float for distances in the air and are transmitted via breathing, talking, singing, sneezing, or coughing [33].

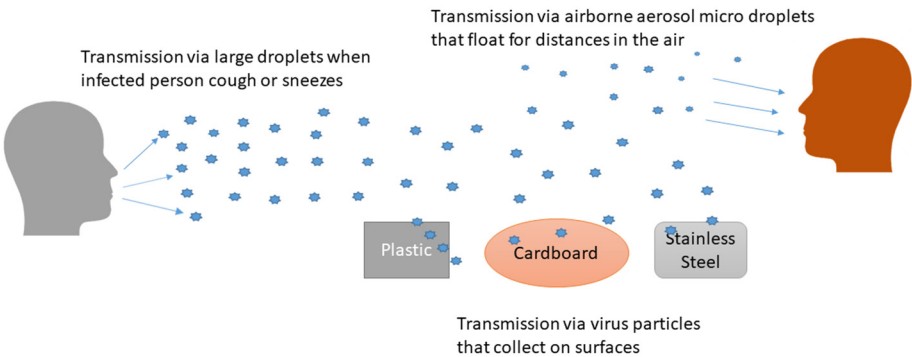

**Figure 2.** Controls should address three ways of virus transmission.

### 3.4.1. Safe Use of Facemasks and Other PPE

Healthcare workers or caregivers are suggested to use N95 or KN95 respirators when providing care for proper protection [39]. Because of the shortage of N95 respirators during the pandemic, the CDC recommended use of cloth masks or disposable surgical masks by the general public as a way to help stop the spread of the virus [40]. Homemade cloth masks are not considered fully effective at preventing the wearer from being exposed to the virus [39–41]. However, they are helpful in containing the wearer's coughs or sneezes as an important preventive measure to protect others around them [41,42]. The crisis for shortage of medical grade effective masks is over, and N95 respirators and respirators complying with international standards, such as KN95, FFP2, FFP3, KN100, and KP95 masks, should be encouraged. They can provide better protection for domestic workers and are available at reasonable prices [43,44].

Several precautions can be taken when wearing a facemask or a cloth mask as follows [34,35]:

1.  Avoid touching face while wearing a mask;
2.  A face mask should be snugly fit over the bridge of the nose and below the chin, but not stretched so tightly that the fabric is flat up against the nose and lips;
3.  Be careful when taking the mask off—do not touch the front of the mask, instead, remove by just touching the straps;
4.  After removing the mask, dispose of it right away if it is disposable, or set it aside to wash after each use if it is a cloth mask;
5.  Make sure to wash hands carefully right after taking the mask off.

### 3.4.2. Safe Use of Chemicals

There is a list of some of the Chemicals (a total of 431 disinfectants listed by the EPA) shared by the EPA that can be used as disinfectants against COVID-19 [34]. Bleach-based products can cause or exacerbate asthma, so if possible, domestic workers using disinfectants should look for products that use hydrogen peroxide instead [14]. Employers can provide non-latex disposable gloves, such as nitrile gloves, for them to use and follow safety precautions while using cleaning chemicals [14,34,35]. OSHA has published guidelines for safe use of chemicals in 2012 [14].

3.4.3. Safety Precautions When Using Cleaning Chemicals [14]

1.   Follow the instructions on product labels exactly.
2.   Provide good ventilation by opening windows or doors.
3.   Never mix cleaning products together or use one product on top of another.
4.   Avoid spraying chemicals into the air or spraying to surfaces; it is better to spray into a cloth and then wipe the surface.
5.   Wash hands thoroughly while cleaning and before leaving the job.

3.4.4. Use of Gloves

Gloves can help limit exposure to the coronavirus from frequently touched surfaces (especially if someone is unable to wash hands). They are also important to protect skin from chemicals in disinfectants while cleaning [14,34].

When using gloves, employees should make sure to follow these tips [14,45]:

1.   Avoid touching your face.
2.   Do not reuse disposable gloves—throw them in the trash after each use.
3.   For disposable gloves, non-latex ones like nitrile gloves are the best.
4.   If using non-disposable gloves (heavy duty rubber or other sturdy material), wash your hands with soap and water with the gloves on before removing them.
5.   Wash your hands thoroughly right after removing gloves.

Health and safety training is an integral part of any prevention plan. Training should be given a priority for employees, especially during the time of pandemic, and it should focus on topics such as basic rights on the job, safe practices, proper PPE use for the job, and addressing new situations during the pandemic [34]. State funds can be devoted to supporting such safety and health training in the languages of the workforce. Domestic workers should be trained on proper safe practices and the use of proper PPE in controlling risk factors associated with the spread of disease [34,45]. Employers should develop safe practices for their domestic workers and have plans to identify signs, symptoms, and risk factors associated with COVID-19 and implement safe hygiene practices at home to prevent the spread of COVID-19 [15,23,34,45].

*3.5. OSHA Guidance on Safe Return of Workers*

On March 2020, OSHA published a report about the guidance on preparing workplaces in COVID-19, which stated how COVID-19 spreads and affects workplaces and health and proposed safe ways to reduce exposure to COVID-19 [34]. The report talks about the Occupational Risk Pyramid for COVID-19, which identifies different job risks. The lower-risk and medium-risk populations include most U.S workers, and high-risk populations include medical transport workers, healthcare delivery, support staff, and mortuary workers, and very high-risk populations include health care workers, lab personnel, and morgue workers. OSHA further lays out safe work practices for each risk groups and included effective control methods—engineering, administrative, and PPE methods—for each risk group [34].

OSHA published guidance on returning to work in June 2020 [35]. In this report, OSHA talked about guiding principles including hazard assessment, hygiene, social distancing, identification, isolation of sick employees, returning to work after illness or exposure, controls, workplace flexibilities, training, and anti-retaliation with details and examples on how to implement and follow each of them in the workplace. This was created as a generalized report for all industries and not particularly created for any specific sectors. OSHA talked about reopening plans in all three phases. Phase 1 reopening plans included telework, social distancing and feasible accommodations for vulnerable populations (sick and elderly). Phase 2 plans included resuming travels for non-essential business but continuing to telework if feasible. Phase 3 reopening plans will allow for resuming unrestricted staffing at worksites. One major point in the report that needs to be mentioned is that OSHA did not recognize cloth masks as PPE but included them as an administrative control. As an explanation of including cloth masks as an administrative

control and not PPE, OSHA report stated that cloth masks do not protect wearers but protect others from the respiratory secretions of the wearer [35].

OSHA updated its advisory guidance in August 2021 on mitigating and preventing the spread of COVID-19 in the workplace for both vaccinated and unvaccinated workers. The guidance, which applies to all industries, encourages employers to mandate vaccinations for employees and implement regular testing requirements for unvaccinated employees. OSHA's latest guidance recommends that fully vaccinated workers in areas of substantial or high community transmission wear masks in order to protect unvaccinated workers. OSHA also recommends that fully vaccinated workers who have close contacts with people who have been diagnosed with COVID-19 wear masks for up to 14 days unless they have a negative coronavirus test at least 3–5 days after such contact. OSHA clarifies recommendations to protect unvaccinated workers and other at-risk workers in manufacturing, meat and poultry processing, seafood processing, and agricultural processing, and provided the latest guidance on K-12 schools and CDC statements on public transit [45].

*3.6. Guidance on Mental Health Protection for Domestic Workers*

Domestic workers find themselves trying to take care of basic needs, such as food and providing for their families. If they do not work, they cannot provide their family with their basic needs; Maslow's Hierarchy of Needs [46] details psychology involved with a person's response to environmental stressors. Maslow's hierarchy of needs covers five tiers in ascending order: physiological needs (food and clothing), safety, love and belonging, esteem, and self-actualization [47]. If the worker's basic needs are not met, they will often not move to the next level, which allows for safety to be a priority. Mental stress, the unknowns from not meeting these basic needs, is coupled with the mental anxiety of working during a pandemic.

Recommendations to reduce mental anxiety would be as follows:

1.  Encourage and provide information regarding vaccination options.
2.  Educate undocumented and documented domestic workers on rights for medical care. Physicians are not required to report undocumented immigrants when providing care.
3.  Allow for sick time if a domestic worker were to get sick or if a positive exposure requires quarantining.
4.  Provide resource contacts to those domestic workers that may also be in charge of their children's remote learning.
5.  Provide local resource contacts to assist with basic needs, including churches, schools, and food pantries.
6.  Open a line of communication.
7.  Provide a consistent pay schedule.

3.6.1. Mental Health Responses and Community Outreach

A study published in Lancet Psychiatry in 2020 reflected on the challenges for mental health that COVID-19 poses and an opportunity to improve mental health services. The interconnectedness of the world has made society vulnerable to this infection, but it also provides the infrastructure to address previous system failings by disseminating good practices that can result in sustained, efficient, and equitable delivery of mental healthcare [48].

Different strategies for community outreach have been used during COVID-19. In the U.S, mental health providers and programs have organized food delivery for vulnerable community members and worked with community leaders to ensure the inclusion of mental as well as physical health concerns in programs [49]. Voluntary sector organizations in many countries have organized emergency funds for struggling people, virtual mutual support meetings, community conversations, and online resources [50]. Some countries have supplemented community support systems by reassigning staff, and volunteers have increased staff numbers [50].

To fill the gaps in face-to-face care during the onset of the pandemic, telehealth was rapidly adopted, with remote video or phone conferencing, online blended or coached therapies, and self-help therapies provided through apps. There is already some evidence of short-term success [51,52], and remote service delivery could have longer-term advantages, especially in countries with low investment in mental health services and low capacity [53].

There are many online and meet up support groups for mental health prevention. The Alcohol, Drug, and Mental Health (ADAMH) group (https://adamhfranklin.org; accessed on 31 October 2021) is a support group for people dealing with substance abuse, alcohol, and mental health issues. The Caregiver Action Network (CAN) (https://www.caregiveraction.org; accessed on 31 October 2021) is the nation's leading family caregiver organization working to improve the quality of life for more than 90 million Americans who care for loved ones with chronic conditions, disabilities, disease, or the frailties of old age. The Triad Mental Health (https://resources.harriscountytx.gov; accessed on 31 October 2021) support group offers support, guidance, and resources to people living with anxiety disorders like panic disorder and social disorder. It supports people living with post-traumatic stress disorder (PTSD) and obsessive-compulsive disorder (OCD). Joining a mental health support group can heal from issues caused by mental health conditions and help to lead a happy and positive life.

### 3.6.2. Opportunities for Jobs

There are an approximate 2.2 million domestic workers in the United States [7]. Many domestic workers in the U.S are undocumented. There are an estimated 3.2 million undocumented workers throughout the United States [54]. One out of five undocumented immigrants works as a landscaping worker, maid or housekeeper, or construction laborer [55]. Undocumented immigrants are not eligible for unemployment, supplemental nutrition assistance program (SNAP) benefits, or access to proper preventative healthcare or other government support programs [56]. Due to these challenges, many undocumented immigrants are forced to work regardless of their safety. Undocumented immigrants are often not tracked and not counted in the reports published by the Bureau of Labor Statistics. There is lack of detailed information available on how many households forgo the need for domestic workers in the U.S. This information, if found, could have been significantly helpful.

According to the U.S Bureau of Labor Statistics (BLS.Gov), total employment is projected to grow from 153.5 million to 165.4 million over the 2020–2030 decade, an increase of 11.9 million jobs. This increase reflects an annual growth rate of 0.7%, which accounts for recovery from low base-year employment for 2020 due to the COVID-19 pandemic and its associated recession. Employment in the leisure and hospitality sector is projected to increase the fastest, largely driven by recovery growth, while the healthcare and social assistance sector is projected to add the most new jobs. Among occupational groups, healthcare support occupations are projected for the fastest job growth. Domestic workers who are in the caregiving role are expected to find jobs as more jobs will be created for them to return to [57].

### 3.7. House Employer's Role in Rehiring Domestic Workers and Practicing Safety and Health to Prevent Spread of Diseases

Employers should keep an open mind when hiring or rehiring domestic workers and not think of domestic workers as a source of spreading viruses [23]. If employers cannot keep an open mind and positive attitude and trust towards their domestic workers, it will make the relation between the employee and their domestics vulnerable [23]. The result will be an added mental stress and the extremely difficult situation may lead to poor quality of work and dissatisfaction. The employer has to step up and make an effective health and safety plan and communicate with their domestic employees to keep their household safe while getting access to the domestic help they need for their family [15,23].

Recommendations for domestic workers and their employers are listed below [23,35,45,58].

To keep domestic workers and family members safe, employers should consider the following recommendations:

1. Minimizing human contact—residents can leave the house for workers like house cleaners, pet sitters, and home repair technicians while the worker inside the home completes the job.
2. Stay in closed room or separate part of the house—to avoid contact and maintain social distancing, residents can stay in a separate area where worker is not expected to clean or perform other work.
3. Pay workers remotely and not in person—this way employers could avoid the human-to-human interaction.
4. Provide personal protective equipment (PPE)—home employers should provide proper PPE (masks and gloves) for the domestic work without any cost to employee and encourage employees to wear proper PPE when possible. According to the CDC, some respirators are designed and tested to meet international standards such as KN95, DL2, DL3, DS2, DS3, FFP2, FFP3, KN100, KP95, KP100, P2, P3, PFF2, PFF3, and R95 equipment, which can give better protection for domestic workers [44].
5. Practice hygiene—frequent hand washing, not sharing food or drinks from the same plates/glass and maintaining social distancing (while taking children or pet to walk) should be practiced.
6. Check for temperature and antibodies—frequent temperature checks for household members and domestic workers can be done as a precaution. Domestic workers should not feel that they are the ones who are checked for health issues, not the family members. Employers need to make sure their employees do not feel that they are looked at as a burden who can be blamed for spreading disease.
7. Open communication between employees and their employers—open communication and trust between employers and their employees are both important. Without proper communication and trust, relations can be difficult. If the employee gets sick, they should be able to stay at home with pay without having to fear the repercussion from the employer. If any of the family members of the employer are sick, the employer should let the employee know and let them stay out of the house or take necessary precautions until the family member gets well.
8. Watch news for changes in incidents (new cases) of coronavirus—if the number goes up for COVID-19 cases in the area, employers should be able to quarantine with their family members and provide the opportunity for their domestic worker to do the same. Without fear of losing their job or wages, domestic workers should be able to quarantine with their own families.
9. Encourage vaccinations—vaccinations will be effective against COVID-19 at a certain level and employers can encourage vaccinations for family members and their domestic workers.

### 3.8. Role of Domestic Workers to Keep Themselves and Their Employer's Family Safe

During rise of COVID-19 cases, it was best to practice social distancing by staying at home [45]. This will be very difficult to do long term. Domestic workers have a great role in cleaning houses thoroughly to limit the spread of infection especially during pandemic. Since communities are slowly starting to open up, domestic workers will be in need to take care of sick/elderly or young children/infants. They will need to start working again [7,15].

To keep themselves and the people they care for safe during the pandemic, domestic workers should consider the following recommendations [15,23,35,45]:

1. Limit visitors: limit visitors to the home/workplace.
2. PPE: all adults who will be in the home while the work is being performed should wear a cloth or disposable surgical mask if they will be in close contact.
3. Limit exposure: domestic workers should limit their own exposure to other people (because the person they care for may be particularly vulnerable to serious symptoms

like children or elders; it is especially important that they reduce their chances of being exposed to the virus, so they do not risk spreading it to their client).

4.  Hygiene: take extra handwashing breaks and wash hands thoroughly.
5.  Change of clothes: domestic workers should change clothes when they arrive at work and when they return home from work.
6.  Medical supplies: work with their employer/client to ensure that the home is stocked with cleaning supplies, over the counter medicines, prescription medicines, and non-perishable foods to minimize trips to the store.
7.  Medical visits: talk with their employer/client about backup plans for medical visits or treatments, in order to limit exposure to coronavirus in healthcare settings.
8.  Back up plans: talk with their employer/client about backup care plans in case either they or their client get sick.
9.  Get vaccinated when possible: domestic workers can choose to get vaccinated to protect themselves and others from COVID-19.

The roles of domestic workers and employers in the time of COVID-19 for protection against the spread of virus are listed in Table 1.

**Table 1.** Role of employers/homeowners and domestic workers in the time of COVID-19.

| Employers/Homeowners Should: | Domestic Workers Should: |
| --- | --- |
| Minimize human contact | Limit visitors to the home |
| Stay in closed room or separate part of the house | Wear PPE |
| Pay workers remotely and not in person | Limit own exposure |
| Provide personal protective equipment (PPE) | Maintain hygiene by taking extra handwashing breaks and wash hands thoroughly |
| Practice hygiene | |
| Check for temperature and antibodies | Change clothes before and after work |
| Keep open communication with their domestic workers | Stock medical supplies and cleaning supplies |
| Watch news for changes in incidents (new cases) of coronavirus | Stock non-perishable foods at home |
| Give sick leave for domestic workers and their families when necessary | Have an open communication with employer/homeowner |
| Give time off with pay to domestic workers if homeowner or any family members are sick and suspected with COVID-19. | Have a back-up plan for online medical visits to health care facilities |

## 4. Conclusions

Domestic workers face an array of challenges including physical and health hazards while working in a home setting [1,2]. The challenges have escalated during the COVID-19 situation with the loss of jobs and loss of income for domestic workers and their families [6]. Rehiring domestic workers can be a complicated process. Domestic workers are usually not trained by employers on safe return policies for work setting. Moreover, some employers can be skeptical in opening their doors to domestic work with the possibility of spreading disease in their homes [23]. Domestic work is essential especially during the time of COVID-19 as domestic workers keep the house clean, take care of the sick and elderly, take care of children while parents head out for work, take care of pets, and perform other essential household duties [6]. Proper plan and open mindedness will be needed on the employer's behalf to reopen their homes for hiring back domestic workers. Domestic workers should also be prepared for the challenge and ready to take safe precautions and follow the best practices to be able to get back to work again [7,15,23].

**Author Contributions:** R.B. participated in design of the study, conception, gathering data, and analyzing and drafting the manuscript. T.B. and J.B. participated in the conception, data collection, and review of the manuscript. All authors have read and agreed to the published version of the manuscript.

**Funding:** This research received no external funding.

**Institutional Review Board Statement:** Not applicable.

**Informed Consent Statement:** Not applicable.

**Data Availability Statement:** Not applicable.

**Acknowledgments:** We are thankful to the domestic workers for their essential work and to the authors who have conducted research on domestic workers safety and health.

**Conflicts of Interest:** The authors declare no conflict of interest.

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
