# Peer review of "Safe Return to Work for Domestic Workers in the Time of COVID-19"

_covid, doi:10.3390/covid1030048_

Round 1
Reviewer 1 Report
This is an interesting article on the safe return to work of house workers in the context of the current COVID-19 pandemic. The following revisions should be performed.
1-use 'COVID-19' and not other forms like 'COVID' 'Covid-19' etc
2-check sentence at line 53
3-Materials and methods: indicate the period range for the literature search
4-Section 3 should be introduced. for example: 3. Results and Discussion 3.1...
5-explain the meaning of the acronym OSHA at its first usage
6-FLSA is very old (1938). What was the minimum wage at those times and what about today?
7-line 110: spread of SARS-CoV-2
8-line 117: explain here why the house workers have less chances to be rehired
9-Fig 1 and Fig 2: are images taken from previous publications? In this case you should ask permission to the authors of such publications. Otherwise, mention in section 2 how you realized images in figure 1 and 2
10-at pag 5 there is a large mention to homemade face masks. In my opinion less importance should be given to these masks as nowadays FFP2 or similars are largely available at reasonable prices.
11-line 243: phrases should be phases
12-line 259: reword 'who have COVID'
13-line 287: shold provide masks, better if FFP2, KN95 (specify)
14-line 156: SARS-CoV-2 as the virus at the origin of COVID-19 should be introduced in the section 1 (Introduction) not here.
Author Response
"Please see the attachment".

Reviewer 2 Report
- In this review, the authors described about the vulnerability of African, American and latino women. How about the Asian? Is there any literature? Lease better to describe it.
- The reviewer noted some abbreviations OSH, NIOSH, EPA websites. Please describe in full name
3.This systematic review is mainly discussed on the issues I Europe, America and Africa. Therefore, it is not generalized for the whole world population. We recommend to add the place Europe, America and Africa? Or better to add the issues and challenges in Asian countries at the revised manuscript.
Author Response
"Please see the attachment"

Reviewer 3 Report
The review article authored by Bardhan, Byrd and Boyd provided steps and recommendation for safe return to work for domestic worker. This review could be potentially interesting to readers of Covid. However I believe it is still lacking a few key elements (1) authors presented weak review of recent literature in regards to covid-19 and (2) most of the outlined recommendation stemmed of OSHA (cf. comprehensive literature review of the field). Additional comments :
Section 3.1: Authors mostly outlined typical challenges faced by domestic worker pre-covid. The literature reviewed were relatively old and add nothing new to the field. The final paragraph in the section should be the start of the section. Authors should review additional literature (e.g. BLS) on what's the current status of hire? how many remains unemployed? how many returned to work?
Section 3.2: Authors highlighted racial vulnerability in this section. I having a hard time to see the connection and the justification of sex and racial vulnerability in safe return to work.
Section 3.3, 3.4, 3.5: Authors mostly outlined OSHA's safety guidance in safe return to work. I feel like this is a missed opportunity for the authors to touch on safe return to work not just physically but mentally. What are the community support groups available? Are there jobs for them to return to? How many households forego the need for domestic workers? Reviewing these data would add a lot of strength to this article.
Reference 4, 12, 33 - incomplete. Bibliography formatting inconsistent.
Author Response
"Please see the attachment"

Round 2
Reviewer 3 Report
Thank you for the revision. The previous comments were adequately addressed, with markedly improved readability.